# Removal of Specular Reflection Using Angle Adjustment of Linear Polarized Filter in Medical Imaging Diagnosis

**DOI:** 10.3390/diagnostics12040863

**Published:** 2022-03-30

**Authors:** Kicheol Yoon, Jaehwang Seol, Kwang Gi Kim

**Affiliations:** 1Medical Devices R&D Center, Gachon University Gil Medical Center, 21, 774 Beon-gil, Namdong-daero, Namdong-gu, Incheon 21565, Korea; kcyoon98@gachon.ac.kr; 2Department of Biomedical Engineering, College of Medicine, Gachon University, 38-13, 3 Beon-gil, Dokjom-ro 3, Namdong-gu, Incheon 21565, Korea; 3Department of Biomedical Engineering, College of Health Science, Gachon University, 191 Hambak-moero, Yeonsu-gu, Incheon 21936, Korea; tjfwoghkd@gachon.ac.kr; 4Department of Health Sciences and Technology, Gachon Advanced Institute for Health Sciences and Technology (GAIHST), Gachon University, 38-13, 3 Beon-gil, Dokjom-ro, Namdong-gu, Incheon 21565, Korea

**Keywords:** diagnosis, camera imaging, specular reflection removal, linear polarized filter, Malus’ law

## Abstract

The biggest problem in imaging medicine is the occurrence of light reflection in the imaging process for lesion diagnosis. The formation of light reflection obscures the diagnostic field of the lesion and interferes with the correct diagnosis of the observer. The existing method has the inconvenience of performing a diagnosis in a state in which light reflection is suppressed by adjusting the direction angle of the camera. This paper proposes a method for rotating a linear polarization filter to remove light reflection in a diagnostic imaging camera. Vertical polarization and horizontal polarization are controlled through the rotation of the filter, and the polarization is adjusted to horizontal polarization. The rotation angle of the filter for horizontal polarization control will be 90°, and the vertical and horizontal polarization waves induce a 90° difference from each other. In this study, light reflection can be effectively removed during the imaging process, and light reflection removal can secure the field of view of the lesion. The removal of light reflection can help the observer’s accurate diagnosis, and these results are expected to be highly reliable and commercialized for direct application in the field of diagnostic medicine.

## 1. Introduction

Modern medicine requires diagnostic imaging and preventive medicine. Most imaging medicine uses X-ray, MRI, CT, ultrasound, microscope, and cameras for precise diagnosis [1]. The precision diagnosis camera enables the real-time observation of lesions [2,3,4,5]. The diagnostic camera adjusts the brightness of the LED to allow imaging in dark tissue [6]. The disadvantage of the camera during the imaging process is the reflected light (specular reflection) from the lesion due to the light of the LED irradiated to the lesion. The light reflection generated by the LED interferes with the observation field of the lesion. In addition, the generation of light reflections prevents accurate diagnosis. Therefore, the emergence of light reflection makes it very difficult to observe the lesion [7].

For tissue diagnosis, if light reflection is created due to the strong light intensity of the LED, the diagnostic field of view is blocked during the monitoring process of the camera. For example, when light reflection generates on the lesion for observation, the lesion cannot be seen, making it difficult to make an accurate diagnosis.

In order to suppress the occurrence of light reflection in the medical diagnosis field, there is a cumbersome need to change the orientation angle of the camera. In addition, the observer faces the cumbersome and complicated process of compensating for light reflection using separate software when shooting is completed. In these methods, it is difficult to expect a rapid diagnosis, and the color of the lesion may be lost in the course of the observer’s work. Then, the patient and the doctor in charge of diagnosis cannot expect accurate diagnosis results. To overcome this problem, several methods for light reflection suppression have been studied [8]. However, research cases are still lacking, and the traditional trick method used in the field of medical diagnosis has been investigated. Most of the medical field has used a method of minimizing light reflection by adjusting the camera’s direction angle [9].

To obtain the polarization effect of the camera, there is a method of irradiating the focal point of the LED in various directions and a method of controlling the beam focal length of the LED [10]. However, it is not easy in the field to adjust the beam irradiation direction angle and focal length of the LED during the diagnosis process [11]. In addition, the best imaging condition for light reflection suppression is to detect specular reflections generated in the spatial conversion process with RGB color functions for the brightness and color of the camera [12]. The detected specular reflection removes the reflection region through deconvolution analysis in the wavelength band. The color image lost due to specular reflection removal restores the lost color through the use of a separate function [12]. This reflected wave cancellation process requires a lot of data and calculation work, which requires complex mathematical operations [12]. The method of collecting frames through continuous shooting requires that the reference image be trained in advance [13]. In this method, the sum and difference are calculated by comparing the interval between the actual shooting frames and the reference training image interval. Then, the reflected light is removed by removing the image of the difference value [13]. However, various cumbersome data collection processes for learning the reference image, lesion data, and normal tissue data follow [14].

There is a method that involves using a polarizing filter in the camera and a method using a sensor that detects only the RGB values of the image by automatically detecting changes in the image [15,16,17,18]. However, the method of applying the polarizing filter reduces the shooting radius due to material loss of the filter and provides a dark image. Moreover, the RGB value detection method does not take into account the expected image loss and distortion due to material loss in the filter [15,16,17,18].

Research is needed to overcome the complex process of data collection and mathematical operation to suppress the light reflection of photographed data. This study is judged to be usefully applied to the field of medical diagnosis. This paper proposes a method for removing light reflection of an imaging camera. In order to remove light reflection, the filter controls vertical and horizontal polarizations through rotation with 90°, which induces the 90° difference from each other. At this time, the filter used is the linear polarized filter. Section 1 introduces the motivation for this study. Section 2 and Section 3 present the research methods and results, and Section 4 a discussion. Lastly, Section 5 concludes the article.

## 2. Analysis for Removal of Specular Reflection

In the diagnostic endoscopy procedure, a camera is inserted into the organ as shown in Figure 1, and the endoscope camera confirms the lesion status of the mucosa. When photographing a dark organ from a picture, the camera surroundings irradiate the LED on the tissue to capture a bright image.

The large problem in the endoscopic camera imaging process is the emergence of light reflection, as shown in Figure 2 [7]. The occurrence of light reflection obstructs the observation field of the lesion, making it difficult for an observer to make an accurate diagnosis. This light reflection controls the polarization direction of the filter by analyzing the characteristics of polarization and Malus’ law [19,20,21]. In consequence, the light reflection can be eliminated.

To remove light reflection, two filters (*E*_1_, *E*_2_) are configured, as shown in Figure 3. In Figure 3, *E*_1_ and *E*_2_ are the first and second polarization filters, and the *E* and H are the electric and magnetic fields of polarization. LPL is linear polarization. *p*_1_, *p*_2_, and *p*_3_ are the positions where the image and light pass through the filter, and *E_s_* is the sum of *E_z_* and *H_z_* (*E_s_* = *E_z_H_z_*). In this case, *E_z_* is vertical polarization and *H_z_* is related to horizontal polarization. Consequently, the image and light are mixed together. *E_o_* is a mixture of image and light with the same horizontal polarization as *E* and *H*. *E_p_* means an image from which the *E_x_*_,*y*_ component (light reflection) is removed and the light reflection in which only the *H_y_* component remains.

In the figure, the light (*E_s_cosθ_s_*) of *p*_1_ incident on the camera through the imaging of the lesion has vertical polarization (*E_y_*) and horizontal polarization (*E_z_*) proceeding in the *x*, *y*, and *z* directions as shown in Equations (1) and (3) (*E_y_* = 1, *H_x_* = 1) is assumed [20,21]. In the formula, *E_z_*_(*y*, *x*)_ means *E_z_* of magnitude in the *y* and *x* directions. *θ* denotes the filter rotation direction angles of *E*_1_ and *E*_2_. In the *z*-direction light (*p*_2_) passing through the polarization filter of *E*_1_, *E_o_*, and *E_s_* are changed to horizontal polarization (*E_y_* = 0, *H_x_* = 0), which is the sum of *E_o_* and *E_s_*, resulting in *E_o_* = *E_s_* = 0 [20,21].
(1)$$E_{z}=E_{y}cos\omega \theta_{t}$$
(2)$$H_{z}=H_{x}cos\omega \theta_{t}$$
(3)$$\sum^{\;}E_{z}=E_{y}+E_{x}=\{\begin{array}{c} E_{z}=E_{z\left(y\right)}e^{j\left(kz-\omega t+\theta \right)}\\ E_{y}=E_{z\left(x\right)}e^{j\left(kz-\omega t+\theta \right)} \end{array}$$

The *z*-direction light (*p*_3_) passing through the filter of *E*_2_ only passes through *E*_0_ in the horizontally polarized state. Then, *E_o_* becomes *E*_0_ ≠ *E_s_*, and *E_o_* becomes *E_o_* = 0. Since the phase of *E_o_* may differ by 0 or an integer multiple of 2π, *E*_0_ proceeds in the *x*-direction as in Equations (4) and (5). All of the polarized waves of *E_o_* moving in the *x*-direction are changed to horizontal polarization (*E_o_* = 0), and the image is changed to a state in which light reflection is removed [21]. Finally, concerning *E*_0_ (*p*_2_), as in Equation (6), vertical polarization and horizontal polarization differ by π in the *z* direction. Eventually, only the horizontal polarization corresponding to π/2 passes through the polarization filter of *E*_2_, resulting in reduced light reflection. An image (*E_p_* of *p*_3_) is provided [21].
(4)$$E_{0}=E_{y}-H_{y}cos\omega \theta_{t}$$
(5)$$\sum^{\;}E_{0}=E_{x\left(y\right)}e^{j\left(kz-\omega t+\theta_{z}\right)}+jE_{y}e^{j\left(kx-\omega t+\theta_{y}\right)}=\left[E_{x\left(y\right)}e^{j\theta_{z}}+jE_{y}e^{j\theta_{y}}\right]e^{j\left(kx-\omega t\right)}=E_{o\left(x,y\right)}e^{j\left(kx-\omega t\right)}$$
(6)$$E_{p}=H_{y}cos\omega \theta_{t}$$
where *p*_1_ is polarized in the vector plane of *x*, *y*, and *z*, and has vertical and horizontal vibration directions (*x*, *y*) at a direction angle (*θ*) of 0°. The polarized waves *p*_1_ of *E_y_* and *H_y_* passing through the first polarization filter *E*_1_ oscillate *p*_2_ in the vertical direction *y*. When the direction angle *θ* of the second polarizing filter *E*_2_ is 90°, only the *E_y_* component passes through the vertical polarized wave of *p*_2_ passing via the second polarizing filter *E*_2_, and the *H_x_* intensity is weakened. The propagation angle of the polarization with respect to *E_p_* can be analyzed using the ABCD matrix shown in Equation (7) and Figure 4 and Figure 5 to reduce the light intensity according to the change in *θ* [21]. If the intensity of light *E*_(*x*, *y*)_ with respect to polarization rotates from *y* to *z* and *θ* is *cos* 90°, light reflection is reduced to the zero (0).
(7)$$E_{p}=\left[\begin{array}{c} E_{x}\\ E_{y} \end{array}\right]=\left[\begin{array}{c} E_{x}e^{j\left(j\theta_{x}\right)}\\ E_{y}e^{j\left(j\theta_{y}\right)} \end{array}\right]=\left[\begin{array}{c} a\\ b \end{array}\right],\;\left|a^{2}\right|+\left|b^{2}\right|=1$$

When *θ* becomes Δ*Φ_i_* (Δ*Φ_i│i = Φz − Φx_* = 0, π/2, π/4, 3π/4, π), the light intensity (*E_o_*_(*x*, *y*)_) changes, as shown in Figure 6.

For example, when the light reflection intensity (*I_ref_*) passes through two filters (*θ_Ep_* = 90° difference) (*E*_1_ = 0°, *E*_2_ = 90°), the light reflection intensity (*I_ref_*) at the *p_3_* position is reduced by more than half. Eventually, the intensity of light reflection (*I_ref_*) is lowered by more than half compared to the intensity of light (*I_Ep_*). The reason for this is that, as in Malus’s law in Equation (8), when the rotation angle of the first filter (*θ_ref_* = 0°) is fixed, when the second filter is rotated at 90° (*cosθ_Ep_* = 90°), the intensity of light reflection (*I_ref_*) becomes 0 [19,20,21].
(8)$$I_{ref}=I_{Ep}cos^{2}\theta_{Ep}=\left(\frac{I_{Ep}}{2}\right)cos\theta_{Ep}{}^{2}=I_{ref}-I_{Ep}=I_{ref}-\left[\left(\frac{1}{3}\right)I_{Ep}\right]=\frac{3(I_{Ep}-I_{Ep})}{3}=\left(\frac{1}{3}\right)I_{Ep}$$

If the *I_Ep_* is 0 mW/cm^2^, the rotation angle (*θ_ref_*) of the *E*_1_ filter is 0°, and the rotation angle (*θ_Ep_*) of the *E*_2_ filter is 90°, as in Equation (9), the light reflection intensity (*I_ref_*) is interpreted again as 50 mW/cm^2^, as given in by Table 1, when *I_Ep_* is 0 mW/cm^2^ (*θ_Ep_* = 90°). At this time, when the rotation angle *θ_ref_* of the *E*_1_ filter is 0°, the rotation angle *θ_Ep_* of the *E_2_* filter corresponds to 0° to 360°.
(9)$$I_{ref}=I_{Ep}cos^{2}\theta_{Ep}=\left(50\times 10^{-3}\right)cos^{2}\left(0{}^{\circ}-90{}^{\circ}\right)=\left(50\times 10^{-3}\right)cos^{2}\left(90{}^{\circ}\right)=\left(50\times 10^{-3}\right){\left(0\right)}^{2}=0\;\mathrm{mW}/{\mathrm{cm}}^{2}$$

When analyzing Equations (1)–(9) and Table 1, since the first polarizing filter (*E*_1_) was fixed at 0°, the second polarizing filter (*E*_2_) was sequentially rotated from 0° to 360°. Similarly, a phenomenon in which the intensity of light reflection decreases and rises is repeated like a sine wave. When the rotation angle *θ_Ep_* of the *E*_2_ filter is 90° or 270°, the intensity of light reflection becomes 0 mW/cm^2^, and a phenomenon in which light reflection is removed arises. For this reason, as shown in Figure 7, when the rotation angle (*θ_Ep_*) is 0°, light (light reflection) passes through the second polarization filter (*E*_2_), and the light reflection intensity (*I_ref_*) has a maximum value. That is, the intensities for *I_Ep_* and *I_ref_* are the same and they have an equilibrium relationship with each other. When the second polarizing filter *E*_2_ rotates from 10° to 80° during the rotation process, *I_Ep_* and *I_ref_* are out of horizontal relationship.

When the rotation axis of the second filter (*E*_2_) reaches 90°, *I_EP_* and *I_ref_* (the difference between *E*_1_ ≠ *E*_2_ = 90° of the filter) intersect with each other, so that *I_EP_* has the maximum value (*I_Ep_* = 1) and *I_ref_* will have the minimum a value of (*I_ref_* = 0). Consequently, the intensity of light reflection is lost, and the image quality is increased. If the rotation angle of the filter deviates from 90° (<100°), *I_EP_* changes to the minimum value (*I_EP_* = 0) and *I_ref_* changes to the maximum value (*I_ref_* = 1). At this time, the *I_EP_* and *I_ref_* will have the same value (*I_EP_* = *I_ref_*). This phenomenon is repeated. To summarize, the method for maximally suppressing light reflection is that when the first filter *E*_1_ is 0°, the second filter *E*_2_ should be 90°. As a result, the polarized wave *p*_3_ that has passed through *E*_2_ outputs only the polarized wave *E_p_* with reduced light reflection. As a result, the intensity of light loses its maximum value, and eventually, the intensity of light starts to decrease slowly, and when the rotation angle *θ_Ep_* is 90° (270°), the intensity of light reflection (*I_ref_*) becomes 0.

## 3. Experiment Composition and Results

In order to obtain the effect of reducing light reflection, the experimental device is configured as shown in Figure 8. The device for the experiment consists of a camera, an LPL filter, and an LED. The method followed to obtain the result uses phantom. The first polarizer filter connected in front of the camera has a rotation angle (*θ*) of 0° and the filter is fixed. However, the filter is in a polarization state. The second filter has a rotation angle (*θ*) of 0° and this filter is unpolarized. The filter connected to the LED changes the rotation angle (*θ*) from 0° to 360°, and the filter is rotated.

The filter for connection to the camera and LED was made using 3D printer technology to make a fixed frame, and the LED and filter are connected in this frame. The installation angle of the camera is 0° and the irradiation direction angle of the LED (the angle between the camera and the LED) is 30°. Moreover, the number of irradiated LEDs is 10. The working distance (WD) between the camera and the LED and the phantom is 14 cm.

The main parameters of the LED (010C3UC020, DAKWANG, Yantai, China), camera (Dr. Cervicam C20, NTL Healthcare, Seongnam, Republic of Korea), and filter used for the experiment are presented in Table 2.

For the use of a linear polarized filter (Schneider-Kreuznach, Schneider (1.25 inch), I55543, Bad Kreuznach, Germany), the wavelength band is 420–750 nm and the diameter of the lens is 27 mm. In addition, the thickness of the filter lens is 0.25 mm, and the transmittance is 34%. Extinction ratio is >10.000:1 and the LPL filter is uncoated.

The experimental results using phantom are shown in Figure 9. From the figure, specific reflection is generated on the phantom. The arising position of the specific reflection is indicated by a yellow arrow. As a result of using the filter, when the rotation angle (*θ*) of the filter connected to the LED is 0°–360°, it can be observed that light reflection is reduced, and the rotation angle (*θ*) is 90°. When it is analyzed, that light reflection is reduced.

The captured image results were simulated using an ROI (region of interesting) program (PyCharm. JetBrains, Prague, Czech Republic), as shown in Figure 10. In the simulation result, when the rotation angle *θ* of the filter is 0°, the light reflection generation position is indicated by a yellow point. In this case, the intensity (intensity of light) corresponds to 0 in the area where light reflection does not occur in the histogram. However, since most of the light reflection is generated to the phantom, the light reflection intensity is no longer 0. Therefore, the intensity of light reflection has a value of 25–294, and the maximum intensity of light reflection is 294. However, when the rotation angle (*θ*) of the filter was 90°, the intensity of light reflection changed to 0.

Figure 11 shows the human tissue test results for the phantom imaging results in Figure 9 for the reliability performance test. For the reliability of the study, this experiment was conducted three times. In order to increase the reliability of the research methodology, direct pictures were taken using my own oral cavity instead of a phantom. In the experimental results, light reflection happens when the LPL filter was not applied, but when the rotation angle (*θ*) of the filter is adjusted from 0° to 360° using the LPL filter, when the rotation angle was 90°, the light reflection was sufficient and could be removed.

In Figure 11, when the LPL filter was used, the phantom, uvula, lingual frenulum, and palatine raphe had light reflections like yellow markers. When the LPL filter (first) is used, the phantom, uvula, lingual frenulum, and palatine raphe, like yellow markers, have light reflections removed so that the image looks clear. When the reproduction test (second and third) was performed to obtain a reliable image to obtain the effect of removing light reflection when the LPL filter is used, the image result of the first LPL filter was similar.

## 4. Discussion

This study proposes a method to efficiently reduce the light reflection of images generated by diagnostic systems, such as microscopes, endoscopes, and cameras.

The mention of the genesis of light reflexes in all tissues is cautious. However, soft fibrous tissues, such as the stomach, colon, and cervix, for endoscopic diagnosis are confirmed in the clinical field as light reflection generates. When the LED is irradiated to the tissue mucosa, light reflection rises according to Snell’s law. Ambient reflection, diffusive reflection, and specular reflection are highly likely to occur because of the difference between the density of the cavity, the tissue density, and the moisture density of the tissue as the cause of light reflection [22].

In the light of the LED irradiated from the cavity, both the absorbed light and the reflected light are generated at the same time because of the density at the water surface interface or the tissue surface interface. The reflected light causes scattering, which prevents monitoring through camera shooting.

In the experimental process, the light reflection intensity is almost identical in the wavelength band of 400–1100 nm. Vertical polarization and horizontal polarization are generated together in the wavelength range. For that reason, it is important to eliminate all vertical polarizations. The wavelength band of 400–1100 nm is a white light source LED. During the endoscopic procedure, white light (LED) is used to brightly image the dark tissue space. Light reflections are severe in white illuminated LEDs (400–1100 nm). In order to reduce light reflection, it is inconvenient to adjust the orientation angle of the camera or adjust the brightness of the LED at the clinical diagnosis site [9]. However, using these methods, it is not easy to obtain clear diagnostic results in the clinical diagnosis process. In particular, if the camera orientation angle is adjusted, the lesion shape observation angle may be changed. In addition, if the brightness of the LED is adjusted, it is observed with a dark background. Accordingly, it is time-consuming and cumbersome to correct the photographing result using an imaging process. For quick and accurate diagnosis, a clear image must be provided, and light reflection must be removed to secure the observation field of the lesion. Accordingly, it is judged that the evaluation of the method of removing light reflection of the camera will contribute a lot to the clinical diagnosis phenomenon.

In this study, a phantom (a silicone material for suturing practice) was used instead of an animal experiment to examine the effect of light reflection removal, and the tissue in Figure 11 was taken using my oral cavity. Consequently, the experiment through oral imaging was repeated three times and the reliability of the results was increased. If a filter made of film material is used, it is judged that the proposed method can be sufficiently commercialized by applying it to a camera shooting system. Thereby, if excellent results are obtained through clinical trials, it is expected that the use value in the field of clinical diagnosis will increase.

## 5. Conclusions

This study proposes a method for removing light reflection in diagnostic imaging systems, such as microscopes, endoscopes, and cameras, and proves the possible results through experiments. The main method for the proposed study is to control the polarization using the rotation angle of the filter. Thus, by controlling the propagation directions of the vertical and horizontal polarizations, the light reflection for the vertical polarization is removed and only the horizontal polarization for the video image can pass through. A phantom was used to obtain the experimental results, and the photographic experiments were repeated three times through the oral tissue to prove the high reliability of results. In consequence, the oral imaging results were consistent with the phantom test results without any light reflection.

When an LPL filter was used, light reflection is generated to the phantom and tissue. However, when the LPL filter (first) was used, the image showed the light reflection removed from the phantom and tissue. Moreover, a reproduction test was conducted to obtain a reliable image, and the image result was similar to the image result of the first LPL filter.

The proposed method eliminates light reflection in real time at the diagnostic imaging site without additional imaging software correction processing, providing excellent imaging. Thus, it is possible to secure a field of view for observation of the lesion through clear imaging results and to obtain fast and accurate diagnosis results. In addition, the advantage of the proposed method can be connected to all cameras, microscopes, and endoscopes regardless of the size of the filter, so it can be applied to clinical diagnosis sites. So, the research method is highly practical. It is expected that this research method can be sufficiently applied to clinical diagnosis sites through product design and clinical trials in the future.

## Figures and Tables

**Figure 1 diagnostics-12-00863-f001:**
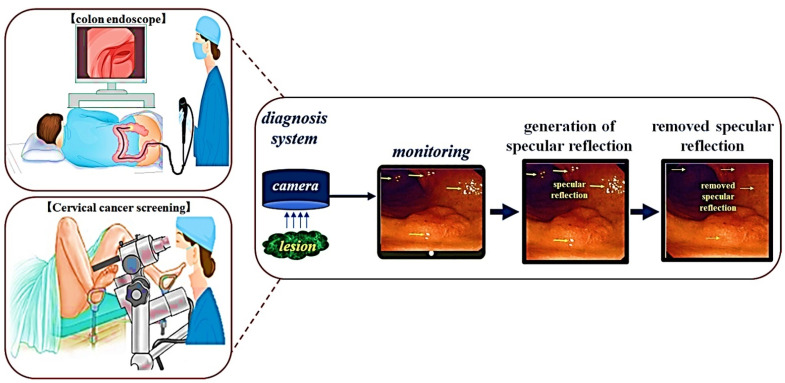
Generation of specular reflection in the diagnosis progress (The occurrence of light reflections from colonoscopy diagnosis and cervical endoscopy. In addition, the phenomenon of obstructing the observation of the lesion due to the occurrence of light reflection).

**Figure 2 diagnostics-12-00863-f002:**
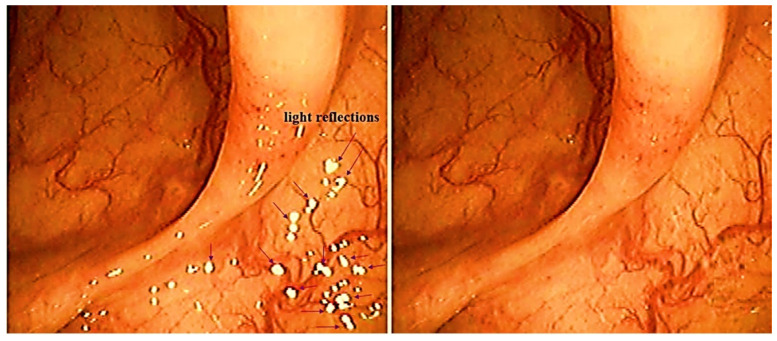
Creation of specular reflection through the LED with colon endoscopy and specular reflection removal.

**Figure 3 diagnostics-12-00863-f003:**
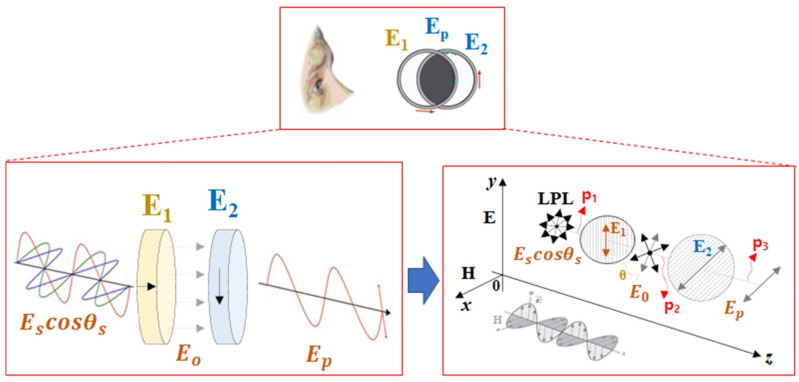
Analysis of specular removal though the linear polarized filtering.

**Figure 4 diagnostics-12-00863-f004:**
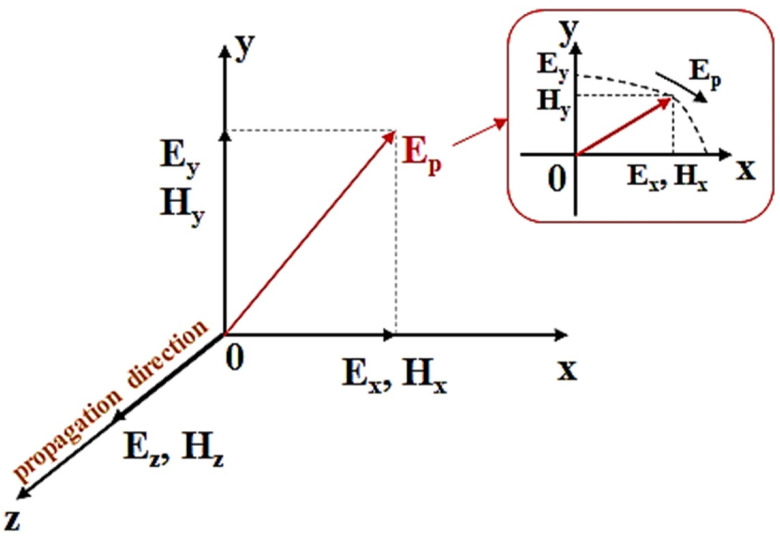
Direction of filter rotation angle (*E*_2_).

**Figure 5 diagnostics-12-00863-f005:**
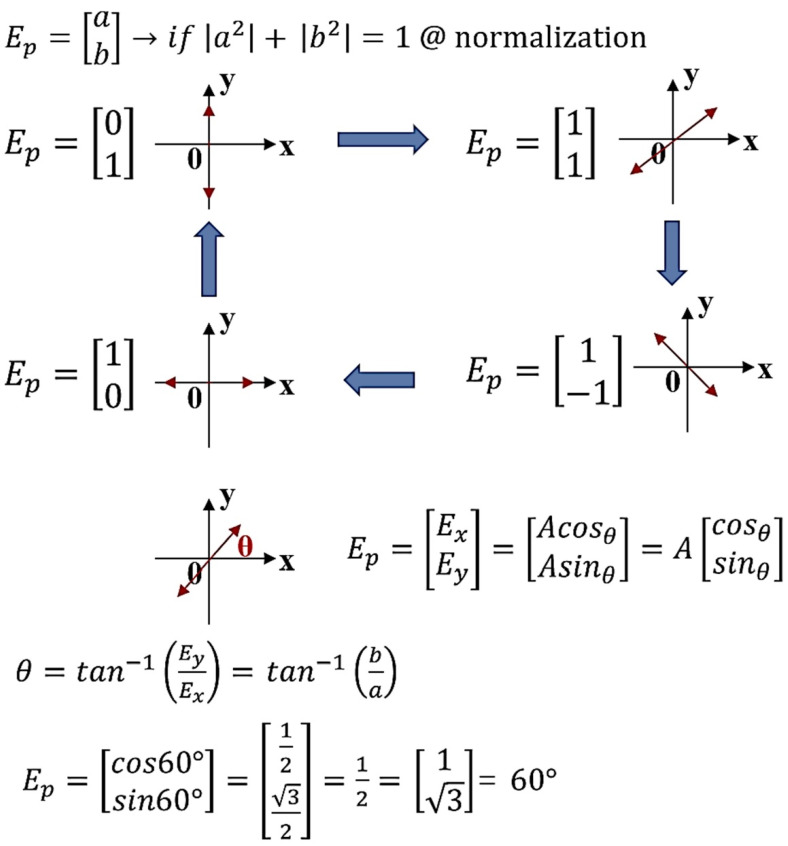
Polarization (specular reflection) propagation direction with respect to the rotation angle of the filter (*E*_2_).

**Figure 6 diagnostics-12-00863-f006:**
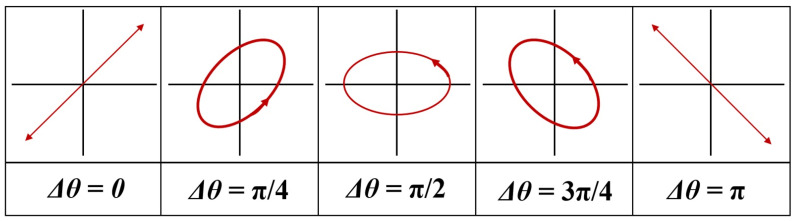
Characteristics of polarization according to the rotation angle of the filter (*E*_2_).

**Figure 7 diagnostics-12-00863-f007:**
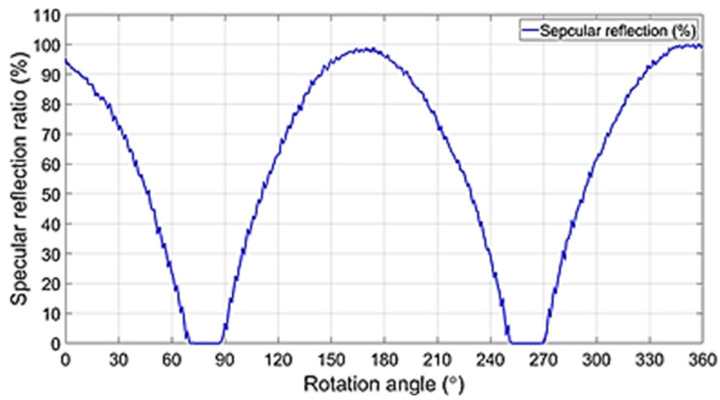
Changes in light reflection intensity according to the rotation axis of the filter using Malus’s laws.

**Figure 8 diagnostics-12-00863-f008:**
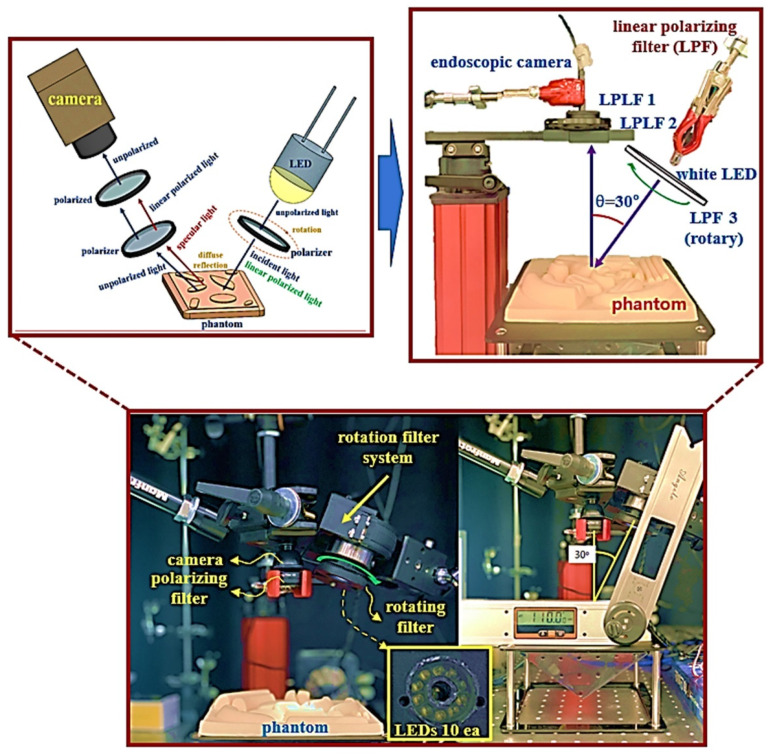
Experimental setup configuration.

**Figure 9 diagnostics-12-00863-f009:**
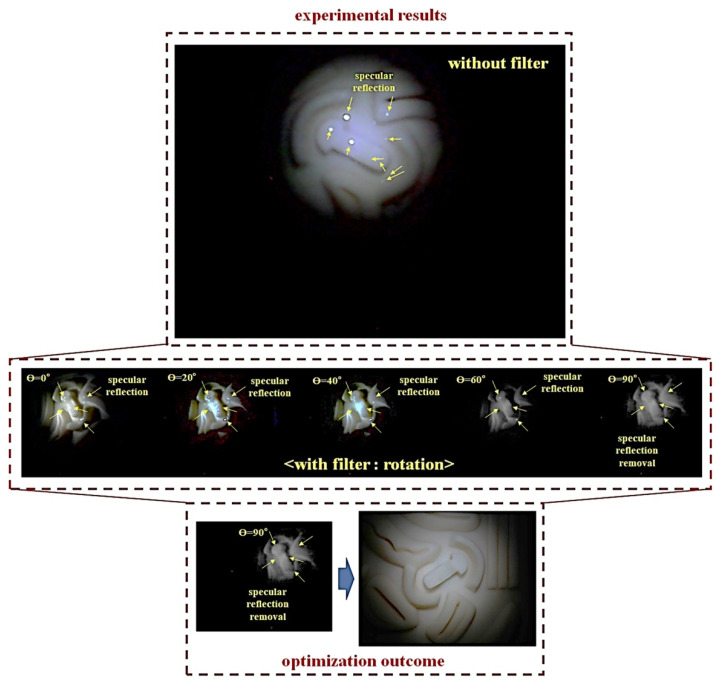
Phantom experiment result for removal specular reflection.

**Figure 10 diagnostics-12-00863-f010:**
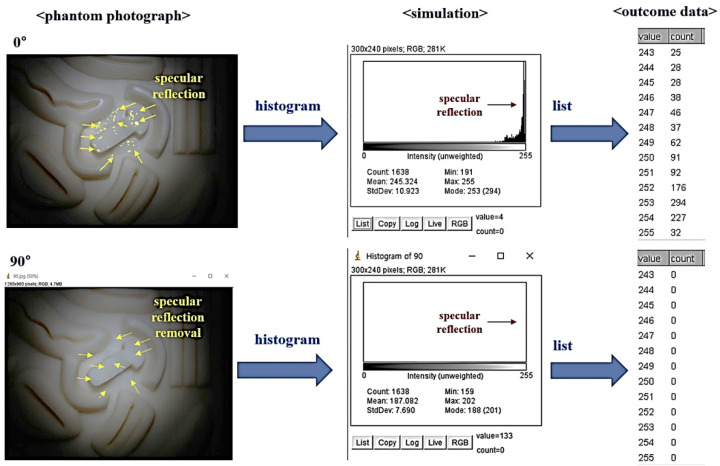
ROI Numerical analysis of Phantom experiment results for specular reflection removal.

**Figure 11 diagnostics-12-00863-f011:**
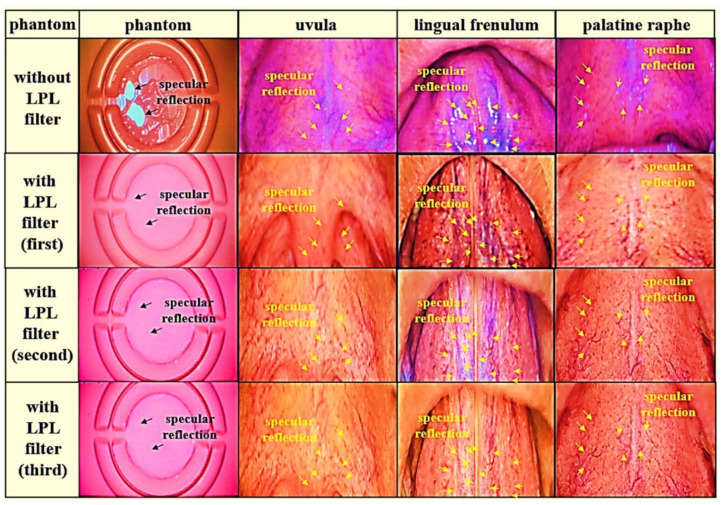
Experimental results of removal specular reflection through oral imaging.

**Table 1 diagnostics-12-00863-t001:** Analysis of changes in light reflection intensity according to the rotation angle of the filter (*E*_2_ @ *E*_1_ = 50 mw/cm^2^).

*E*_2_ Rotation Angle of the Filter (θ_Ep_)	Light Reflection Intensity [mW/cm^2^]	*E*_2_ Rotation Angle of the Filter (θ_Ep_)	Light Reflection Intensity [mW/cm^2^]
0°	50.0	210°	37.5
30°	37.5	240°	12.5
60°	12.5	270°	0.00
90°	0.00	300°	12.5
120°	12.5	330°	37.5
150°	37.5	360°	50.0
180°	50.0		

**Table 2 diagnostics-12-00863-t002:** Experimental device module parameters.

Performance (@ LED)	Parameter	Performance (@ Camera)	Parameter
model	LED (010C3UC020)	model	SJ-8200
wavelength, *λ* [nm]	465–470	sensor	CMOS
output power [mW]	50	Resolution [P]	1920
current [mA]	20	pixel size [M_pixel_]	2.0
voltage [V]	2.5	frame rete [fps]	30
beam angle of radiation, *θ* [deg]	40	focal distance	5 mm–infinity
luminous intensity [mrcd]	2500	view angle [deg]	60°

## Data Availability

The data presented in this study are available upon request from the author.

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
