# Peer review of "Removal of Specular Reflection Using Angle Adjustment of Linear Polarized Filter in Medical Imaging Diagnosis"

_diagnostics, 2022, doi:10.3390/diagnostics12040863_

Round 1

Reviewer 1 Report

In this manuscript, authors have addressed an efficient way for removing specular reflection from the tissue surface, which is quite important in the context of medical imaging application such as endoscopy, colonoscopy, or any medical cavity imaging contexts. The approach proposed by the authors is very simple, straightforward and is widely adopted in optical engineering. However, accounting to its significance in suppressing strong specular reflection from tissue surface which saturates the imaging pixels and or obscures the lesion/or abnormality in the tissue structures under investigation, their contribution in this manuscript is worthy . Authors have nicely and systematically presented the contents with appropriate illustrations. The manuscript is overall written well, however, need changes based on the following comments.

  1. The malus law is a fundamental law which provides the theoretical value of the intensity of light variation with respect to the orientation of polarizer (w.r.t analyzer). Also, the polarization equation used are very basic and could be found in the basic optics textbooks. These equations are not authors contributions. Authors need to strictly use references from these equations are adopted.
  2. I found similar graphics showing orientation of polarization direction and its components in other textbooks, and online media. Authors should present their own sketches and images in the manuscript. This should be in accordance with the copyright rules and regulations. This is applicable to Fig 1 and Fig 8 as well
  3. The specular reflection could also be avoided by the changing illumination angle and/or changing the detection path. These approaches are also recently demonstrated in retinal OCT imaging contexts which further allowed for quantification of reflectivity (https://doi.org/10.1117/1.JBO.24.6.066011, https://doi.org/10.1038/s41598-021-95320-z). This is another best example of showing importance of removing of specular reflection in medical imaging. Authors may mention this approach with references in the introduction section.
  4. Do you think, if a polarized source is used, could we remove the polarizer used at the illumination path? Will it be economical if we do that way?
  5. Are these polarizers used dependent on wavelength of illumination? Usually, polarizer is designed w.r.t wavelengths used. For medical imaging applications such as endoscopy or oral examination, white light is used? In such cases, how do you choose polarizers?
  6. Will the polarizers used affect the field of view?
  7. Could authors detail, what causes the specular reflection from tissues? Is it caused by Fresnel’s reflections? Will all biological samples have these issues?
  8. Which polarizer has authors used for their applications? Need to provide all details including model name, size, manufacturer etc

Author Response

Point 1: The malus law is a fundamental law which provides the theoretical value of the intensity of light variation with respect to the orientation of polarizer (w.r.t analyzer). Also, the polarization equation used are very basic and could be found in the basic optics textbooks. These equations are not authors contributions. Authors need to strictly use references from these equations are adopted.

Response 1: References [19]-[21] were added and further analysis was performed to solve the problem. Please refer to equation (8).

Point 2: I found similar graphics showing orientation of polarization direction and its components in other textbooks, and online media. Authors should present their own sketches and images in the manuscript. This should be in accordance with the copyright rules and regulations. This is applicable to Fig 1 and Fig 8 as well.

Response 2: Figure 1 and Figure 8 were redrawn using the “hand made” method.

Point 3: The specular reflection could also be avoided by the changing illumination angle and/or changing the detection path. These approaches are also recently demonstrated in retinal OCT imaging contexts which further allowed for quantification of reflectivity (https://doi.org/10.1117/1.JBO.24.6.066011, https://doi.org/10.1038/s41598-021-95320-z). This is another best example of showing importance of removing of specular reflection in medical imaging. Authors may mention this approach with references in the introduction section.

Response 3: Thanks for the advice. Thanks for introducing the reference of best practices. Added [9] on lines 54-57 (gray) in the introduction.

Point 4: Do you think, if a polarized source is used, could we remove the polarizer used at the illumination path? Will it be economical if we do that way?.

Response 4: The vertical polarization must be removed to match the horizontal polarization generated by the LED source and the horizontal polarization being captured by the camera. Also, all vertical polarization should be removed. Just adjust the rotation angle of the filter to remove the vertical polarization from the camera. However, in order to remove the vertical polarization from the LED, a polarizing filter must be used in the LED source.

Point 5: Are these polarizers used dependent on wavelength of illumination? Usually, polarizer is designed w.r.t wavelengths used. For medical imaging applications such as endoscopy or oral examination, white light is used? In such cases, how do you choose polarizers?.

Response 5: In the experimental process, the light reflection intensity is almost identical in the wavelength band of 400-1100 nm. Vertical polarization and horizontal polarization are generated together in the wavelength range. Therefore, it is important to eliminate all vertical polarizations. The wavelength band of 400-1100 nm is a white light source LED. During the endoscopic procedure, white light (LED) is used to brightly image the dark tissue space. Light reflections are severe in white illuminated LEDs (400 nm -1100 nm).

Thanks for the good comments. Added above to discussion (283-288 lines, gray) Will the polarizers.

Point 6: Will the polarizers used affect the field of view?

Response 6: I have missed something in the introduction, and thank you very much for pointing it out. The following sentences were reflected in the introduction (43-46. lines: Sky blue). thank you.

For tissue diagnosis, if light reflection occurs due to the strong light intensity of the LED, the diagnostic field of view is blocked during the monitoring process of the camera. That is, when light reflection occurs on the lesion for observation, the lesion cannot be seen, making it difficult to make an accurate diagnosis.

Point 7: Could authors detail, what causes the specular reflection from tissues? Is it caused by Fresnel’s reflections? Will all biological samples have these issues?

Response 7: Corrected and supplemented. Chap. 4. Discussion

Check out Lines 272-278 (green).

The mention of the occurrence of light reflexes in all tissues is cautious. However, soft fibrous tissues such as the stomach, colon, and cervix for endoscopic diagnosis are confirmed in the clinical field as light reflection occurs.

When the LED is irradiated to the tissue mucosa, light reflection occurs according to Snell's law. Ambient reflection, diffusive reflection, and specular reflection are highly likely to occur because of the difference between the density of the cavity and the tissue density and the moisture density of the tissue as the cause of light reflection [b]. In the light of the LED irradiated from the cavity, both the absorbed light and the reflected light are generated at the same time because of the density at the water surface interface or the tissue surface interface. Therefore, the reflected light causes scattering, which prevents monitoring through camera shooting.

Point 8: Which polarizer has authors used for their applications? Need to provide all details including model name, size, manufacturer etc.

Response 8 : The main model names and specifications for the experiment are as follows.

Also see lines 225-226, 229-230, 240-241 (yellow).

  • LED: 010C3UC020, DAKWANG, China)
  • camera : Dr. Cervicam C20, NTL Healthcare, Republic of Korea
  • filter : (Schneider-kreuznach, Schneider (1.25 inch), I55543 Bad Kreuznach, Germany
  • imaging capture program: PyCharm. JetBrains, Czech Republic.

Reviewer 2 Report

  1. I suggest a new title for the paper, maybe not repeating the terms “polarization” and “polarized”: Removal of specular reflection using angle adjustment of linear polarized filter in medical imaging diagnosis.
  2. The English spelling of the manuscript should be improved. For example, the author uses constantly the term “therefore”, sometimes in two consequent sentences. The English must be conveniently improved.
  3. Introduction:
    1. First paragraph: the last two sentences have a similar structure and words, why not join?
    2. The authors repeat the word “occurrence” and “therefore” several times. Please review.
    3. Line 55: …there are a method of… Correct form: there is a method of.
    4. Line 68: “Therefore, the reflected light is removed by removing the image of the car”. I do not understand this sentence. Please, introduce some contextualization.
    5. Figure 1: consider a more complete caption of the figure. Include “specular reflection”.
  4. Section 2
    1. Line116, there is a mistake (Ey). =1.
    2. Line 125 and 127 Eo (I think the authors write with o instead of 0).
    3. Figure 5 - there is a mistake in the equation related to the example for 60 degrees. There is an extra “=” at the end of the equation.
    4. Line 162/163: “… is given in Equation (8), the ratio of transmitted amplitude to cos becomes θ.” What do the authors mean? Explain and rewrite this sentence.
    5. Line 168/169: “…the light reflection intensity (Iref) is It can…” Review this sentence and the term “It”.
  5. Section 3
    1. Line 212: This sentence does not make sense: “When the configuration method of Figure 8 (left) is applied, Figure 8 (right) is the configuration device for the actual experiment.”
    2. Line 228: “From the picture, the image taken before using the filter had specular reflection like a yellow marker.” I think the authors want to say “… as highlighted with the yellow arrows.”
    3. Figure 11: How do you explain the differences between the different images? What is the advantage of applying 2 LPL filters instead of 2 (even 1)? These results should be conveniently analyzed and discussed.
  6. Conclusions: review lines 285/286. “… and repeated experiments were repeated 3 times…”

Author Response

Point 1: I suggest a new title for the paper, maybe not repeating the terms “polarization” and “polarized”: Removal of specular reflection using angle adjustment of linear polarized filter in medical imaging diagnosis.

Response 1: Thanks for the new title suggestion.

The title has been edited as follows.

Title before change: Removal of specular reflection using polarization direction angle adjustment of Linear Polarized Filter in medical diagnostic imaging camera

Title after change: Removal of specular reflection using angle adjustment of linear polarized filter in medical imaging diagnosis

Point 2: The English spelling of the manuscript should be improved. For example, the author uses constantly the term “therefore”, sometimes in two consequent sentences. The English must be conveniently improved.

Response 2: Intercessory words have been removed by using various words. thank you.

Point 3: Introduction:

  1. First paragraph: the last two sentences have a similar structure and words, why not join?
  2. The authors repeat the word “occurrence” and “therefore” several times. Please review.
  3. Line 55: …there are a method of… Correct form: there is a method of.
  4. Line 68: “Therefore, the reflected light is removed by removing the image of the car”. I do not understand this sentence. Please, introduce some contextualization.
  5. Figure 1: consider a more complete caption of the figure. Include “specular reflection”.

Response 3: a. When I checked, I noticed that there were a lot of duplicate words. So I made a correction (lines 89-90, gray). Thanks for the comments.

  1. Intercessory words have been removed by using various words. thank you.
  2. The semantic form is introduced as follows. Please refer to Lines 58-60 (sky blue).

To obtain the polarization effect of the camera, there are a method of irradiating the focal point of the LED in various directions and a method of controlling the beam focal length of the LED [9].

  1. “car” is a typo and is exactly the difference value. I made a correction, please refer to the yellow color on lines 72.
  2. Fixed with specular reflection.

Point 4 : section 2

Response 4: Line116, there is a mistake (Ey). =1.

  1. Line 125 and 127 Eo (I think the authors write with o instead of 0).
  2. Figure 5 - there is a mistake in the equation related to the example for 60 degrees. There is an extra “=” at the end of the equation.
  3. Line 162/163: “… is given in Equation (8), the ratio of transmitted amplitude to cos becomes θ.” What do the authors mean? Explain and rewrite this sentence.
  4. Line 168/169: “…the light reflection intensity (Iref) is It can…” Review this sentence and the term “It”.

Point 5 : section 3

  1. Line 212: This sentence does not make sense: “When the configuration method of Figure 8 (left) is applied, Figure 8 (right) is the configuration device for the actual experiment.”
  2. Line 228: “From the picture, the image taken before using the filter had specular reflection like a yellow marker.” I think the authors want to say “… as highlighted with the yellow arrows.”
  3. Figure 11: How do you explain the differences between the different images? What is the advantage of applying 2 LPL filters instead of 2 (even 1)? These results should be conveniently analyzed and discussed.

Response 5: a. Edited to “right”. thank you..

  1. Yes. All right. A yellow marker on the Without filter indicates that light reflection has occurred. And with filter means that light reflection is eliminated.
  2. Figure 11 has been improved. And I added content. Before and after using the filter is recorded in detail. See lines 262-268 (green) for the sentence in Figure 11.

Point 6 : Conclusions: review lines 285/286. “… and repeated experiments were repeated 3 times…”

Response 6 : I have complemented the results through review. Please refer to Lines 314-321 (Green).

Round 2

Reviewer 2 Report

Some of my previous comments and suggestions were addressed in this new version of the manuscript. However, some of them were not reviewed, namely:

  1. Introduction:
    1. First paragraph: the last two sentences have a similar structure and words, why not join?
    2. The authors repeat the word “occurrence” and “therefore” several times. Please review.
    3. Line 55: …there are a method of… Correct form: there is a method of.
    4. Figure 1: consider a more complete caption of the figure. Include “specular reflection”. (I refer to the caption of Figure 1, not to the figure itself)
  2. Section 2
    • Figure 5 - there is a mistake in the equation related to the example for 60 degrees. There is an extra “=” at the end of the equation.
  3. Section 3
    1. Line 212: This sentence does not make sense: “When the configuration method of Figure 8 (left) is applied, Figure 8 (right) is the configuration device for the actual experiment.” (I refer to the sentence, not to the figure)
    2. Line 228: “From the picture, the image taken before using the filter had specular reflection like a yellow marker.” I think the authors want to say “… as highlighted with the yellow arrows.”
  4. Conclusions: review lines 285/286. “… and repeated experiments were repeated 3 times…”

Author Response

Point 1: I Introduction:

  1. First paragraph: the last two sentences have a similar structure and words, why not join?

Response a : I revised the structure of sentences with last two sentnces. please refer to 83-89.

  1. The authors repeat the word “occurrence” and “therefore” several times. Please review.

Response b : I refined the grammar as a whole and minimized the words "occurrence" and "therefore”. Duplicate words were minimized.

  1. Line 55: …there are a method of… Correct form: there is a method of.

Response c : Thank you for pointing out the sentence correctly. I made a correction. Please refer to line 58 (yellow).

  1. Figure 1: consider a more complete caption of the figure. Include “specular reflection”. (I refer to the caption of Figure 1, not to the figure itself)

Response d : I didn't understand the major revision. However, I finally understood (minor revision). Thank you very much for pointing it out for a better result. I made a correction. Please check the title and caption of Figure 1. If you make a mistake, please give me an example to help me understand. Then I will do my best to correct it. Thank you.

Point 2: Section 2

  • Figure 5 - there is a mistake in the equation related to the example for 60 degrees. There is an extra “=” at the end of the equation.

Response 2: I understood your question and understood your intentions. I removed the 60 degree part from Figure 5. It was judged that it was meaningless in the process of revising the thesis. I also modified the esophagus for Ep. Also, the title of Figure 5 has been revised to "special reflection" (yellow) (formerly light reflection). In addition, please refer to lines of 149-150 (yellow).

Point 3: Section 3

Point a : Line 212: This sentence does not make sense: “When the configuration method of Figure 8 (left) is applied, Figure 8 (right) is the configuration device for the actual experiment.” (I refer to the sentence, not to the figure)

Response a: I deleted the following sentence from the document. It is meaningless for me to judge. “When the configuration method of Figure 8 (left) is applied, Figure 8 (right) is the configuration device for the actual experiment. From the figure, the experimental configuration method is to hold the filter, LED, and cameras using clamp devices, and a phantom is installed on the floor.”

Point b : Line 228: “From the picture, the image taken before using the filter had specular reflection like a yellow marker.” I think the authors want to say “… as highlighted with the yellow arrows.”

Response b: I revised the sentence as follows. Also, please refer to line 230-232 (gray).

“The experimental results using phantom are shown in Figure 9. From the figure, specific reflection occurred in the phantom. The occurrence position of the specific reflection is indicated by a yellow arrow”.

Point 4 : Conclusions: review lines 285/286. “… and repeated experiments were repeated 3 times…”

Response 4: I have considered the sentence. And I made some modifications to the sentence. Please refer to Lines 312-313 (yellow).